# The Fergana Valley Is an Isolate of Biodiversity: A Discussion of the Endemic Herpetofauna and Description of Two New Species of *Alsophylax* (Sauria: Gekkonidae) from Eastern Uzbekistan

**DOI:** 10.3390/ani13152516

**Published:** 2023-08-04

**Authors:** Roman A. Nazarov, Timur V. Abduraupov, Evgeniya Yu. Shepelya, Mariya A. Gritsina, Daniel A. Melnikov, Matthew D. Buehler, Jack D. Lapin, Nikolay A. Poyarkov, Jesse L. Grismer

**Affiliations:** 1Zoological Museum, Lomonosov Moscow State University, B. Nikitskaya 2, 125009 Moscow, Russia; evgeniyashepelya@gmail.com; 2Institute of Zoology Academy of Science of the Republic of Uzbekistan, Bagishamol Str., 232b, Tashkent 100053, Uzbekistan; timur.abduraupov@gmail.com (T.V.A.); mgritsina@gmail.com (M.A.G.); 3Zoological Institute of the Russian Academy of Sciences, Universitetskaya nab. 1, 199034 St. Petersburg, Russia; melnikovda@yandex.ru; 4Department of Biological Sciences and Museum of Natural History, Auburn University, Auburn, AL 36849, USA; mbuehle3@gmail.com; 5Biodiversity Institute, Dyche Hall, University of Kansas, 1345 Jayhawk Blvd., Lawrence, KS 66045, USA; 6Faculty of Biology, Lomonosov Moscow State University, 119991 Moscow, Russia; n.poyarkov@gmail.com; 7Department of Biology, La Sierra University, Riverside, CA 92505, USA

**Keywords:** reptiles, conservation, taxonomy, Gekkonidae, *Alsophylax*, Fergana Valley, endemic species, cryptic diversity

## Abstract

**Simple Summary:**

This study was carried out in one of the most densely populated and geographically isolated regions in the Republic of Uzbekistan, the Fergana Valley. The Fergana Valley has the highest level of endemic biodiversity in Uzbekistan (and one of the highest in Central Asia), and the habitats of these endemic species are rapidly being developed for agricultural purposes. Given this development, the remaining areas of habitat are not being adequately protected. The main goal of this study was to obtain up-to-date data on the distribution and abundance of five endemic reptile species in the remaining isolated and undeveloped habitats across the Fergana Valley. One of the most important achievements was the discovery of two unique and new micro-endemic species of gecko genus *Alsophylax*, which are described herein. These results elevate the number of endemic species in the Fergana Valley and further highlight the urgent need to create state-protected areas of habitat with IUCN I and II protection status for the remaining areas of suitable habitat, which is currently not available.

**Abstract:**

The high level of endemism in Fergana Valley has been well documented in numerous studies for various groups of animals and plants. In a relatively small area, there are 45 endemic plant species, five endemic insect species, and five endemic reptile species. In surveying this area for data on distribution, abundance, acoustics, and genetic samples for species of reptiles, we discovered two new species of gecko from the genus *Alsophylax*. Phylogenetic analyses of mitochondrial DNA sequences indicate the relatives of these new species are the even-fingered gecko, *Alsophylax pipiens*, and the southern even-fingered gecko, *Alsophylax laevis*, located hundreds of kilometers to the northwest and southwest of the Fergana Valley. The threats to these new endemic species are significant given the amount of continued agricultural development that involves new territories previously considered “unsuitable” for any species of significance that is leading to the further reduction in, fragmentation of, and degradation of the remaining natural ecosystems in the Fergana Valley. The conservation of these rare and locally endemic species depends directly on the readiness of the state to create areas with IUCN I and II protection. The many studies documenting levels of endemism, along with the data published in this study, are the basis for the justification for state-protected areas in the Fergana Valley.

## 1. Introduction

The Fergana Valley is located in the eastern region of Uzbekistan and represents an ancient, isolated ecoregion with unique flora and fauna, much of which is endemic to the valley. In a relatively small area, the Fergana Valley contains several dozen endemic plant species, five endemic species of insects, and five endemic reptile species, most of which are associated with the sandy massifs and foothill habitats in the western and southern areas [1]. With the level of aggressive agricultural development and effects associated with climate change, each of these endemic species is facing its own set of environmental threats. For example, Strauch’s toad agama (*Phrynocephalus strauchi* Nikolsky, 1899), Said-Aliev’s toad-head agama (*Phrynocephalus saidalievi* Sattorov, 1981), Fergana sand racerunner (*Eremias scripta pherganensis* Szczerbak et Washetko, 1973), and Strauch’s even-fingered gecko (*Alsophylax loricatus* Strauch, 1887) are all sandy massif specialists, and all five species fall into the “risk zone” [2]. These historically undisturbed sandy massifs, which comprise one of the key habitats for endemic species of reptiles in the Fergana Valley, have been reduced to 175.71 sq. km, and there are currently only 13 isolated plots of habitat, which now occupy an area of 0.46–58.66 sq. km [3].

### 1.1. The Current Status of Conservation in Fergana Valley

Agriculture and textile development has a long history in the Fergana Valley causing the gradual reduction in native habitats. The greatest reduction occurred in the early 20th century due to textiles and vegetable farming, leading to the development of the flatter areas such as sandy habitats. These sandy habitats represent the remains of a vast desert complex in the Fergana Valley originating from the Syrdarya River and are quite old, allowing for adaptation and speciation in these unique microhabitats. The rapid growth of the population of Uzbekistan [3] led to large portions of these habitats being plowed for development. Simultaneously, the rise in population also led to the construction of many agricultural fields, fish ponds, and houses, further fragmenting sandy habitats into sand islands, and preventing migration between adjacent populations. Additionally, excessive watering of adjacent agricultural developments has had a negative impact on existing habitat, resulting in a change to the hydrological regime [3]. This increase in groundwater creates more vegetation growth in these sand habitats, reducing the open cover dune habitat. This reduction in open dune habitat increases interspecific competition between the endemic sand-dune-adapted species and surrounding non-sand-dune-adapted species, making conservation efforts ineffective [4,5]. The problem of preserving sand massif habitats in the Fergana Valley is further complicated by the fact that establishing the correct status and protection regime for these unprotected areas in Uzbekistan is technically quite difficult.

Until 2023, conservation of these endemic reptile species and their habitats was hampered by the fact that, in the Fergana Valley, there were no pre-existing areas with IUCN I and II protection status. The existing natural state monument, “*Yazyavan sands*” [1,2], and other natural district-level monuments do not meet the requirement for higher-level biodiversity conservation [6]. However, since 2020, the project “Preservation of Key Natural Complexes in the Fergana Valley (Republic of Uzbekistan)” through the Michael Succow Foundation, with support from the World Wildlife Fund (WWF), Critical Ecosystem Partnership Fund (CEPF), and Central Asian Desert Initiative (CADI) project, has been garnering more attention for the conservation of the Fergana Valley. Additionally, the project “Maintenance and Conducting Selective Records of Rare and Endangered Species of Vertebrate Animals of the Fergana Valley”, carried out by the “Cadaster and Cadastre of Rare Vertebrate Animals” at the Institute of Zoology of the Academy of Sciences of the Republic of Uzbekistan, has been documenting new populations of species in the Fergana Valley since 2020. In 2019, our team conducted a National Geographic Explorers expedition to survey reptiles in Uzbekistan including the Fergana Valley, and our team returned for multiple expeditions in the Fergana Valley between late 2019 and late 2022. Our team assessed the current state of the remaining preserved isolated habitat sites; as a result, we were able to update distribution and population density data for many reptile species in Uzbekistan. Most of our surveys were focused on the endemic and endangered species in Fergana Valley. Moreover, during these surveys, we discovered two new micro-endemic habitat specialist species of the small body-sized gekkonid genus, *Alsophylax*.

### 1.2. Current Taxonomy of the Gecko Genus Alsophylax

*Alsophylax* Fitzinger, 1843 originated from the description of *Lacerta pipiens* by Pallas [7] and was subsequently placed in its own genus, *Alsophylax*, by Fitzinger [8]. Since then, *Alsophylax* has received little attention due to the fact that its species are very small in size and have what can be perceived as a conserved morphology across the distribution of the genus. As a result, the taxonomic history of this group has been extremely complicated and confounding. Fortunately, there have been detailed descriptions of useful characteristics for describing new species in recent works [1,9,10,11]. Currently, *Alsophylax* comprises six species distributed in Central Asia and southern Russia (*A. laevis* Nikolsky, 1907; *A. loricatus* Strauch, 1887; *A. pipiens* (Pallas, 1827); *A. przewalskii* Strauch, 1887; *A. szczerbaki* (Golubev & Sattarov, 1979); *A. tadjikiensis* Golubev, 1979) [1]. To date, no study has investigated the interspecific relationships within *Alsophylax*, and even its phylogenetic position among other gecko genera remains poorly understood, further exemplifying their understudied nature [12,13,14].

In this study, we provide the most complete phylogeny for *Alsophylax* on the basis of the mitochondrial barcoding gene cytochrome oxidase I, and we include expanded sampling for species complexes such as *A. laevis* and *A. pipiens.* We used this phylogeny along with a dataset of morphological characteristics to demonstrate that these new populations of *Alsophylax* from the Fergana Valley are distinct species and clearly demonstrate the need for further work in Uzbekistan and the Fergana Valley to find other undescribed *Alsophylax* species. Lastly, we provide a discussion on the history of herpetological work in the Fergana Valley, highlighting its endemic reptile species and providing updated distribution and population density assessments of these endemic reptiles.

## 2. Materials and Methods

### 2.1. Population Density and Field Observations

We sampled individuals using the line transect survey method at each sand massif habitat island. During the field observations, we observed 1288 individuals of seven species along the total route length of 210 km. Diurnal species were estimated from routes with varying width along the transect. We recorded the perpendicular distance from the transect line to each individual reptile that was recorded to calculate the transect width We carried out nocturnal sampling using headlamps at a fixed width along the transect. When surveying *Teratoscincus*, we also used the counting of the reddish eye reflections with some specimens detectable at quite far distances up to 100 m or more. However, we understand that using a wide census band can underestimate occurrence data. Therefore, we limited it in accordance with the terrain and vegetation cover of the area. The accounting results, average detection distance ($\bar{y}$) of the species and effective width of the census band (*B*), were calculated. We calculated population density (*D*) per hectare (*ha*) using the following formula:$$D=\frac{N}{2BL};\;B=\frac{\pi }{2}\;\bar{y};\;\bar{y}=\frac{1}{n}\;\sum_{i=1}^{n}y_{i},$$
where *N* is the total number of individual reptiles, *n* is the number of individuals with measured detection distances, and *L* is the route length. Comparative analysis of the results obtained using various route accounting methods proved its high accuracy [15]. This method has been successfully employed in multiple studies to survey desert reptile species [16,17,18]. We calculated mean values of the population density along with estimation of their error (*M* ± *m*) according to the results at several approximately equal route sections.

We conducted daytime surveys in sunny weather with low wind and no precipitation. We started recording data coinciding with the lizards’ maximum activity, and finished when they shifted to shelters to escape the heat during the maximum daytime temperature. We also recorded substrate temperature.

### 2.2. Molecular Data and Phylogenetic Analyses

Our dataset included 46 samples belonging to all six *Alsophylax* species, including the two new species from the Fergana Valley. This also includes one *Altiphylax* species from Central Asia, which we hypothesize is an *Alsophylax*, and one sample of *Mediodactylus kirmanensis* (Nikolsky, 1900) from southern Iran as an outgroup. All analyzed materials are listed in Appendix B.

We extracted total genomic DNA from ethanol-preserved muscle or liver tissues using standard phenol–chloroform extraction procedures [19] followed by isopropanol precipitation. We amplified a fragment of the cytochrome oxidase I (COI) gene with maximal length of 655 bp. This mitochondrial marker is widely used for barcoding in vertebrates [20,21] and has also proven to be useful for species identification in reptiles [22,23,24]. Primers used both for PCR and sequencing were the VF1-d (5′–TTC TCA ACC AAC CAC AAR GAY ATY GG–3′), VR1-d (5′–TAG ACT TCT GGG TGG CCR AAR AAY CA–3′) [25], RepCOI-F (5′–TNT TMT CAA CNA ACC ACA AAG A–3′), and RepCOI-R (5′–ACT TCT GGR TGK CCA AAR AAT CA–3′) [24]. We sequenced fragments in both directions for each sample, and a consensus sequence was generated. We used 25 µL reactions for PCRs with approximately 50 ng of genomic DNA, 10 pmol of each primer, 15 nmol of each dNTP, 50 nmol of additional MgCl_2_, Taq PCR buffer (10 mM Tris-HCl, pH 8.3, 50 mM KCl, 1.1 mM MgCl_2_, and 0.01% gelatine), and 1 U of Taq DNA polymerase. We used the following PCR conditions: an initial denaturation step at 95 °C for 3 min; five cycles at 95 °C for 30 s, annealing at 45 °C for 1 min, and extension at 72 °C for 2 min; 35 cycles at 95 °C for 30 s, annealing at 48–51 °C for 1 min, extension at 72 °C for 2 min, final extension of 5 min at 72 °C, and storage at 4 °C. We loaded PCR products onto 1% agarose gels, stained with GelStar gel stain (Cambrex), and visualized in a Dark reader transilluminator (Clare Chemical). If results were satisfactory, products were purified using 2 µL, from a 1:4 dilution of ExoSapIt (Amersham), per 5 µL of PCR product prior to cycle sequencing. The 10 µL sequencing reaction included 2 µL of template, 2.5 µL of sequencing buffer, 0.8 µL of 10 pmol primer, 0.4 µL of BigDye Terminator version 3.1 Sequencing Standard (Applied Biosystems), and 4.2 µL of water. The sequencing reaction was 35 cycles of 10 s at 96 °C, 10 s at 50 °C, and 4 min at 60 °C. We purified cycle sequencing products using ethanol precipitation. We carried out sequence data collection and visualization on an ABI 3730xl automated sequencer (Applied Biosystems). Resulting sequences are deposited in GenBank under accession numbers (see Appendix B).

All Sequences were aligned manually using BioEdit Sequence Alignment Editor 5.0.9 [26]. The final alignment used for all phylogenetic analysis contained 654 bp of COI gene for all samples. MODELTEST v.3.06 [27] was used to estimate the optimal evolutionary models to be used for phylogenetic analysis. The preferred model was (GTR + G), as suggested by the Akaike information criterion (AIC). Confidence in tree topology was estimated with bootstraps in RAxML 8.1.1 [28]. We considered values above 85% as strong support and rather than choose an arbitrary number of bootstrap replicates to run we use internal metrics of the to allow the analysis to stop itself when sufficient replicates have been run.

### 2.3. Morphological Data and Statistical Analyses

We analyzed 310 specimens of all six recognized species of the genus *Alsophylax* for 22 morphological characteristics, including 14 metric and eight meristic characteristics. We chose these morphological characters on the basis of previous studies on *Alsophylax* [9], as well as additional new characteristics that were identified for this study. All specimens used were from Zoological Museum of Moscow State University (ZMMU), Zoological Institute of St. Petersburg (ZISP), Institute of Zoology National Academy of Science, Ukraine (IZ NAS), and Institute of Zoology Academy of Sciences of the Republic of Uzbekistan (IZ) (Appendix B).

Morphological measurements were taken with a digital caliper to the nearest 0.1 mm and included the following list of characteristics (14 in total): snout–vent length (SVL) measured from tip of snout to vent; tail length (TL), measured from vent to tail tip; head length (HL), distance between retroarticular process of jaw and snout tip; head width (HW), measured at the widest point of the head; head height (HeadH), measured from top of occiput to underside of jaws; orbital diameter (OrbD), diameter of orbit; snout-to-eye distance (SnEye), measured between anterior most point of eye and tip of snout; eye-to-ear distance (EyeEar), distance from anterior edge of ear opening to posterior corner of eye; humeral length (LS), measured on ventral surface of arm from base of axilla to posterior margin of elbow while forelimb bent by 90° at elbow; forearm length (ForeaL), measured on dorsal surface while forelimb bent by 90° at the elbow from the posterior margin of elbow to wrist; femur length (FemurL), measured on the ventral surface of the leg from base of femur to knee while hindlimb bent by 90°; crus length or tibia length (CrusL), measured on ventral surface of the of knee to the base of the heel while hindlimb bent by 90°; length of finger IV (LD4A), length of free distal phalanx of forth finger, without claw; length of toe IV (LD4P), length of free distal phalanx of forth toe, without claw.

The following characteristics (eight in total) were examined: number of scales around midbody including ventral and dorsal scales (Sq); number of scales along midbody from mental shield to anterior edge of cloaca (SLB); number of supralabials (SL); number of infralabials (IL); number of scales along middle of head, between eyes (I); number of subdigital lamellae under fourth finger (LF 4); number of subdigital lamellae under fourth toe (LT 4); number of postcloacal spurs on base of tail on both sides (Spurs); number of precloacal pores (PP).

We investigated morphometric differences between species of *Alsophylax* and the new species described herein. We log_10_-transformed all meristic measurements to normalize the data normality and increase the homogeneity of variance. We adjusted morphometric measurements to remove the effects of body size variation [29]. We use the allometric formula: Xadj = log(X) − b[log(BL) − log(BLmean)].We conducted a discriminant function analysis (DFA) to characterize the morphological variation between one new population of *Alsophylax* from the Fergana Valley and *A. laevis* and *A. pipiens*.

## 3. Results

### 3.1. Population Density and Distribution

Through literary sources, as well as the results of our surveys of the Fergana Valley from 2019 to 2022, we noted 27 species of reptiles, which accounts for 43.5% of all reptile species known from Uzbekistan. Eight (30%) of the 27 Fergana Valley species are considered rare and listed in the Red Book of the Republic of Uzbekistan, five species (18.5%) are listed in the IUCN Red List, and three (11%) are CITES-protected species. Lastly, there are seven (26%) species that are micro-endemic species. The data from our surveys were analyzed and used to make Table 1, which lists all surveyed locations with an estimated average density of each endemic reptile species.

### 3.2. Molecular Analysis and Morphology

All phylogenetic analyses support the monophyly of *Alsophylax* (Figure 1). These analyses demonstrate that both *A. pipiens* and *A. laevis* are polyphyletic, with significant genetic structure within each of these wide-ranging species complexes. These analyses also indicate that both of these new *Alsophylax* populations from the Fergana Valley (Figure 2) are represent unique evolutionary lineages, sister to different clades of either *A. pipiens* or *A. laevis.* All sequences have been deposited on Genbank and their accession number can be found in Appendix C.

Taxonomy

Family Gekkonidae

Genus *Alsophylax* Fitzinger, 1843

Both new species belong to the genus *Alsophylax* according to the following morphological characteristics:

(1) Elongate body with relatively short limbs; (2) non-segmented tail approximately the same length as the body; (3) relatively small and roundish head with a short rostral part; (4) no femoral pores and 6–8 precloacal pores; (5) small, roundish and flattened dorsal tubercles are present in some representatives of this genus, or dorsal scales can be smooth and homogeneous in another species.

*Alsophylax ferganensis* sp. nov. (Figure 3 and Figure 4a)

Holotype. Adult male ZMMU Re-17532 (field ID RAN 4358) collected from urban-type settlement Shorssu (N 40.244792 E 70.821582), Uzbekistan District, Fergana Region, Uzbekistan by Timur Abduraupov and Roman Nazarov in 2021 (Figure 5a–e).

Paratypes. Four males UZZI RE–19077, UZZI RE–19078, ZMMU Re-17537, ZMMU Re-17538 (field ID RAN 4360, RAN 4363, RAN 4370, RAN 4371) and seven females UZZI RE–19075, UZZI RE–19076, Re-17533, Re-17534, Re-17535, Re-17536, Re-17539 (field ID RAN 4359, 4361, 4362, 4364, 4365, 4369, 4372) data same as the holotype (Figure 5f, Table 2).

Diagnosis. *Alsophylax ferganensis* sp. nov. is tentatively a sister to clades A of *A. pipiens* and *A. laevis* (Figure 1) and morphologically closer to *A. laevis* (Table 3). *Alsophylax ferganensis* sp. nov. sp. nov. can be distinguished from *A. laevis* by a smaller maximum body size (SVL_max_ 31.5 mm versus 38.7 mm) and relatively narrow head, as well as elongated limbs and dorsal patterns with well-defined nuchal loop, and relatively narrower dark transverse bands with approximately equal interspaces in between versus wide transverse patterns and narrow interspaces in *A. laevis* (Figure 6). Caudal margins have dark transverse bands that are wavy. Dorsal scales are flat, smooth, and roundish, without enlarged dorsal tubercles (Figure 7).

The main features characterizing and distinguishing it from all of other species are as follows: a maximum SVL of 31.7 mm; 7–8 infralabials; 5–6 supralabials; 1–2 pairs of roundish postmentals, first pair in a broad contact; one nasal scale; 18–22 scales between the orbits of the eyes; 83–97 longitudinal ventral scales from postmentals to cloaca; 44–51 scales along the midline around the body; 13–15 subdigital lamellae on the fourth finger; 14–18 subdigital lamellae on the fourth toe; males have 6–7 precloacal pores on enlarged scales; precloacal pores on females are absent or 7–8 enlarged perforated scales can be present; 2–3 cloacal spurs on each side; with the subcaudal scales, the central line has slightly enlarged roundish plates (Table 3).

Description of holotype. This is an adult male SVL 29.6 mm with a small teardrop shaped head (HL/SVL 0.27), smoothly passing into the neck. The rostral part is roundish and short (SnEye/HL 0.37), more elongated than the occiput (SnEye/EyeEar 1.11). The ear opening is small and roundish. There is a relatively large eye (Orb D/HL 0.25). The distance between posterior margin of the eye and ear (EyeEar/HL 0.33) exceeds the diameter of the orbit of the eye. Along the midline of the rostrum, a longitudinal depression is noticeable. The nostril is surrounded by the first supralabial, rostral, and enlarged nasal scale. Seven supralabials and five large infralabials of a rectangular shape are present, which decrease in length in the caudal direction. A large pentagonal mental shield borders on the first labial and two rounded postmentals. There are 18 round–oval scales between the eyes. The upper part of the head is covered by round, small granular scales which are enlarged in the area between the orbits. Large rounded gulars gradually pass into ventral large flat ventral scales (Figure 5b–e).

The body is elongated and slightly flattened, with no lateral folds. The dorsal scales are relatively small compared to the ventral; slightly larger and smaller scales are evenly distributed among them. No enlarged dorsal tubercles. A total of 86 scales are present from postmentals to cloaca, along with 44 around the midbody. Seven elongated, rectangular, perforated precloacal scales form an inverted V-shaped row. The opening of one pore occupies half of the total area of the precloacal scale.

The forelimbs are slender, and the humeral length slightly exceeds the size of the forearm (ForeaL/LS 0.9). The hindlimb is elongated, stronger in structure than the front. The thigh and lower leg are of the same length (CrusL/FemurL 0.1). The forelegs are dorsally covered with smooth juxtaposed scales, while convex granular scales exist on the ventral side of the upper arm, overlapping each other on the forearm. The dorsal surface of the hindlimb is covered by the scales similar to the dorsal ones, in contrast to enlarged scales on the ventral side, smooth and imbricated. The fingers are slender and long. From above, the metacarpus and metatarsus are covered with scales passing from the forearm and lower leg, respectively; those from below the palm and sole are covered with small swollen rounded granules, and the lower surface of the fingers is covered with transverse subdigital lamellae. The fourth digit of the hindlimb is the longest, with 14 slightly swollen subdigital lamellae.

The tail is 36.9 mm longer than the body (SVL/TL 0.8). The base of the tail is slightly swollen; on each side, there are two pairs of postcloacal spurs. The roundish subcaudal plates on the medial line of ventral side of the tail are 1.5–2 times larger than the surrounding scales.

Coloration. The dorsum of the body, head, and limbs are light beige. The pattern on the dorsum is formed by 5–7 transversely dark brown bands with uneven edges; the interspaces between bands is approximately equal to or slightly greater than the width of the bands. There are 7–10 transverse dark stripes on the tail, about the same number as the width of the gaps. Patterns on the head are not pronounced, whereby there is a narrow nuchal loop. The dorsal surface of the limbs is covered with transverse dark bands and irregularly shaped spots. The ventral surface of the body is white.

Distribution. *Alsophylax ferganensis* sp. nov. is only known from the type locality on the border with Kyrgyzstan, on the southern edge of the Fergana Valley (Figure 2). The distribution of *A. ferganensis* sp. nov. is probably limited to this small mountain range, and a preliminary estimation of the available habitat indicates there is less than 50 km^2^. During field observations, another similar population was found on Pap Adyrs, which is located on the opposite side of the Fergana Valley. This isolated population is located 75 km from the type locality of *A. ferganensis* sp. nov., and its taxonomic status remains unclear.

Habitat and natural history. Clay-variegated canyons with sandstone outcrops represent the habitat of *A. ferganensis* sp. nov. (Figure 4b). The vegetation in these habitats consists of various shrubs and other plants. Most of the geckos were found on open hills with puffy cracked soil and at the bottom of the canyon along the sides.

Comparison. All representatives of the genus *Alsophylax* are divided into two groups—one dominant group has enlarged dorsal tubercles (*A. loricatus*, *A. pipiens*, *A. przewalskii*, *A. szczerbaki*) and the other group does not (*A. laevis*, *A. tadjikiensis*). *Alsophylax ferganensis* sp. nov. belongs to the second group and can be distinguished from *A. loricatus*, *A. pipiens*, *A. przewalskii*, and *A. szczerbaki* by granular homogeneous dorsal scales without enlarged tubercles (Figure 6). *A. ferganensis* sp. nov. sp. nov. is morphologically closest to *A. laevis* but differs further by smaller maximal body size, relatively elongate limbs, and a relatively smaller head that is not sharply detached from the body; dorsal patterns are relatively narrower dark transverse bands with approximately equal interspaces in between, in contrast to the well-defined nuchal loop in *A. laevis* with wide transverse dorsal patterns and narrow interspaces. *Alsophylax ferganensis* sp. nov. differs from *A. tadjikiensis* by the following features: 6–8 precloacal pores in new species versus 5–7 pores in *A. tadjikiensis*; transversal dorsal bands versus pattern-less dorsum; bright yellowish or orange coloration of tail in *A. tadjikiensis*.

Etymology. This species is named after the Fergana Valley where it is endemic.

*Alsophylax emilia* sp. nov. (Figure 3 and Figure 8a)

Holotype. This is an adult male ZMMU Re-17544 (field ID RAN 4917) collected from Uzbekistan, Fergana Valley, Kokand region, in the vicinity of Jildalisoy reservoir, N 40.891349, E 70.884367, by Roman Nazarov, Evgeniya Shepelya, and Mariya Gritsina on 17 June 2022 (Figure 9a).

Paratypes. The adult male UZZI RE–19079 (field ID RAN 4918) collected in the same place as a holotype and ZMMU Re-17545 (field ID RAN 4924) collected from Uzbekistan, Fergana Valley, Namangan region, vicinity of Yartepa village, N 41.067757, E 71.412176, on 18 June 2022 by Roman Nazarov, Evgeniya Shepelya and Bogatova Polina (Table 4).

Diagnosis. *Alsophylax emilia* sp. nov. is a sister to the clade containing lineages of *A. pipiens, A. laevis*, and *A, ferganensis* sp. nov. (Figure 1)*. Alsophylax emilia* sp. nov. is morphologically closest to the *A. pipiens* (Figure 3; Table 3) and can be distinguished by the following features: enlarged dorsal tubercles flat roundish smooth and larger than surrounding scales by no more than 2.5 times; head shape relatively more massive and sharply delimited from the body; 7–9 precloacal pores in V-shaped line; not elongated limbs; 5–6 wide transversal dorsal bands and narrow interspaces, no contrasting nuchal loop, and wide transverse patterns. The main distinguishing features of the new species are the following: maximum SVL of 35.0 mm; maximum TL 40.2 mm; 8/8 infralabials; 6/7 supralabials; two pairs of small roundish postmentals, which contact mentals; one nasal scale; 22 scales between the orbits; 97–104 longitudinal ventral scales from postmentals to cloaca; 59–65 scales along the midline around the body; 11–12 subdigital lamellae on the fourth finger; 13–14 subdigital lamellae on the fourth toe; 2–3 cloacal spurs on each side; medial line of subcaudal scales formed by noticeably enlarged plates, 2–3 times larger the surrounding scales (Table 4).

Description of holotype. This is an adult male, SVL 35.0 mm, TL 40.2 mm, with a relatively large head shape (HL/SVL 0.28) that tapers into the neck and body. The rostrum is rounded and relatively elongate (Snyer/HL 0.31). The distance between rostral and occipital parts is approximately the same (SnEye/EyeEar 1.03). The ear opening is small and oval. The eye is relatively small (Orb D/HL 0.22). The distance between the posterior margin of eye and anterior margin of ear (EyeEar 2.9) exceeds the diameter of the orbit (Orb D 2.15).

The nostril is surrounded by the first supralabial, rostral, and one enlarged nasal scale. There are eight supralabials and six large infralabials rectangular in shape, which decrease in size posteriorly. A large mental shield with roundish caudal edge borders with the first labial and four small and rounded postmentals. There are 22 granular scales between the eyes. The upper part of the head is covered by rounded scales which are enlarged on the sides of rostrum and between the orbits. Large, rounded gulars gradually pass into ventral large flat smooth juxtaposed scales (Figure 9).

The body is elongated and slightly flattened, with no lateral folds. The dorsal scales are heterogenic. Among small roundish scales, evenly distributed enlarged dorsal tubercles do not form regular rows (Figure 9d). Enlarged dorsal tubercles are no more than 2.5 times larger than surrounding scales. In the longitudinal medial row from postmentales to cloaca, there are 97 scales, along with 65 scales around the midbody. There are eight precloacal pores, located on elongated rectangular precloacal scales in an inverted V-shaped row (Figure 9). The opening of the pore is large, occupying more than half of the total area of the precloacal scale. Between the row of precloacal pores and cloaca, there are five enlarged scales.

The forelimbs are slender, and the humeral length slightly exceeds the size of the forearm (ForeaL/LS 0.9). The hindlimbs are elongated, stronger in structure than the front. The thigh (6.5 mm) and lower leg are the same length. The dorsal surface of forelimbs and hindlimb is covered with smooth juxtaposed and imbricated scales, similar in size to the dorsal ones. The fingers are slender and not elongate (LD4A/SVL 0.07). From above, the metacarpus and metatarsus are covered with scales passing from the forearm and lower leg, respectively; those from below the palm and sole are covered with small swollen rounded granules, and the lower surface of the fingers is covered with transverse subdigital lamellae. The fourth digit of the hindlimb is the longest, with 13 slightly swollen subdigital lamellae.

The tail is 40.2 mm longer than the body (SVL/TL 0.9), cylindrical, rounded in cross-section, and not segmented. The base of the tail is slightly swollen; on the sides, there are two pairs of postcloacal spurs. The medial row features enlarged subcaudal plates.

Coloration. The background of the dorsum, head, and limbs are light beige (Figure 9d). The dorsal pattern is formed by six wide transversal dark brown bands with uneven edges; the interspaces between bands are approximately equal to or less than the width of the bands. There are 11 transverse dark bands on the tail. Patterns on the head are not pronounced, with a nuchal loop, dark brown strips on the sides of rostrum between eye, and supralabial plates. Moreover, no contrasting dark roundish spot is present on the tip of the rostrum. The dorsal surfaces of the limbs have transverse dark irregularly shaped spots (Figure 9). The color of the venter is white.

Distribution. *Alsophylax emilia* sp. nov. is known from a few localities on the northwestern edge of the Fergana Valley (Figure 2). Specimens were found at the three localities on pap foothills (adyrs): vicinity of Jidalisai N40.891349; E70.884367; vicinity of Uygursay (N40.95126; E71.01833); 10 km northwest of Turakurgan village (N41.067757; E71.412176); one locality in the Namangan region, vicinity of Yartepa vill. N 41.067757, E 71.412176. We hypothesize that the distribution of *A. emilia* sp. nov. is probably broader in the Fergana Valley, but future studies are needed to clarify the distribution of this species in the surrounding regions.

Habitat and natural history. The habitat at all known localities of *Alsophylax emilia* sp. nov. was similar and made of clay canyons with outcrops, with sparse vegetation consisting of various shrubs (Figure 8b).

Etymology. This species is named in honor of Soviet and then Uzbek herpetologist Emilia V. Vashetko (24.04.1940–07.11.2022) for her great contributions to the study of the herpetofauna of Uzbekistan and surrounding countries.

Comparison. *Alsophylax emilia* sp. nov. belongs to the group of *Alsophylax* with heterogeneous dorsal scales and enlarged dorsal tubercles, which includes *A. loricatus*, *A. pipiens*, *A. przewalskii,* and *A. szczerbaki* (Table 3). *Alsophylax emilia* sp. nov. is morphologically closest to *A. pipiens*, from which it can be distinguished by the smaller dorsal tubercles on new species, which is are no more than 2.5 times smaller than surrounding scales; the medial line of subcaudal scales is formed by noticeably enlarged plates, which are 2–3 times larger than surrounding scales; there are 7–9 precloacal pores versus 8–13 in *A. pipiens* (Figure 8 and Figure 9). *Alsophylax emilia* sp. nov. differs from *A. loricatus*, *A. przewalskii*, and *A. szczerbaki* by having smaller and smooth dorsal tubercles which are sporadically distributed across the back, versus large, strongly keeled trihedral dorsal tubercles organized in regular longitudinal rows; there are smaller scales on the head (22 scales between the eyes versus 8–15 in all three species mentioned above); dorsal patterns consist of transversal dark bands versus unclear dorsal patterns with longitudinal elements (Figure 6).

*Alsophylax emilia* sp. nov. further differs from *A. tadjikiensis* by the following characteristics: heterogeneous dorsal scalation versus homogeneous dorsal scalation; 7–9 precloacal pores in new species versus 5–7 pores in *A. tadjikiensis*; transversal dorsal bands versus pattern-less dorsum and bright yellowish or orange coloration of tail in *A. tadjikiensis*. *Alsophylax emilia* sp. nov. differs from *A. ferganensis* sp. nov. as follows: a larger body size with a maximum SVL of 35 mm versus 30.3 mm; heterogeneous dorsal scalation versus homogeneous dorsal scalation; elongate fore- and hindlimbs on *A. ferganensis* sp. nov.; transversal enlarged subcaudal plates versus roundish polygonal slightly enlarged subcaudal scales along the midline in *A. ferganensis* sp. nov. (Figure 10).

## 4. Discussion

### 4.1. Taxonomy of Alsophylax and Conservation in the Fergana Valley

The genetic and morphological data in this study support the hypothesis that these new populations of *Alsophylax* represent undescribed new species, and that they currently appear to be endemic to the Fergana Valley of Uzbekistan (Figure 2 and Figure 10). Given that these new species tentatively belong to separate species groups within *Alsophylax* suggests that the Fergana Valley may have been colonized by at least two different *Alsophylax* lineages (Figure 1). These results further highlight the age and complex geological processes that have helped generate the unique biodiversity in this isolated valley. Additionally, these new species elevate the number of endemic reptiles to the Fergana Valley and Uzbekistan. Lastly, we provided data on population densities for all endemic reptile species found in the Fergana Valley (Table 1; Appendix A) indicating that not all these species respond to habitat degradation in the same manner. We discuss these results in conjunction with the history of previous herpetological studies in the Fergana Valley, and the future of conservation and species discovery in the area.

### 4.2. The History of Herpetological Study in Central Asia and the Fergana Valley

The history of studies on reptiles inhabiting the territory of Uzbekistan, as was the case with other vertebrates in Central Asia, began with the expedition of E. A. Eversmann and K. Pander from Orenburg to Bukhara (October 1820–April 1821). The few and occasional collections of E. A. Eversmann from these areas were transferred to the University of Berlin, processed by Prof. M. Lichtenstein, and published in 1823 as an appendix to the work of E. A. Eversmann [30].

−H. C. Lichtenstein published a list of 18 species of reptiles (including five new ones) and two species of amphibians [31].−A. Lehmann in July 1841 crossed the Kuvan-Darya and the Kyzylkum desert, and then visited Samarkand and Bukhara. From there, he traveled to the mountains to Iskenderun and went to Orenburg, Kazakhstan [32].−Brandt [33] and Nikolsky [34,35,36,37,38] expanded the known number of species in the area including two species of amphibians and 22 species of reptiles.−A. P. Fedchenko in 1871 worked in the Fergana Valley and visited Kokand, Andijon, Namangan, Vuadil, Uch-Kurgan, and Besh-Aryk. The works by Fedchenko [39,40,41] included only part of the materials he collected in Central Asia and the Fergana Valley. The remaining collections were mainly used by A. M. Nikolsky and A. A. Strauch in their works.−Strauch published a list of geckos from Central Asia, compiled their characteristics, and described a number of new species in Central Asia. Of greatest interest are descriptions of the Turkestan thin-toed gecko, Transcaspian bent-toed gecko, and Strauch’s even-fingered gecko from Uzbekistan and other republics of Central Asia [42].−Nikolsky published “Reptiles and amphibians of the Turkestan general governorship” and gave a complete literary summary at that time [34]. The author processed the materials of A. P. Fedchenko and others who collected in Central Asia. The paper provides information on the distribution of seven species of amphibians, three species of turtles, 42 species of lizards, and 28 species of snakes. Of these, more than half were collected in Uzbekistan [34].−Bogdanov published his work “Fauna of the Uzbek SSR. Amphibians and reptiles” [30] and provided all the data that he collected during his time doing field work in the Namangan region during 1949, 1950, 1954, and 1955.−E. V. Vashetko and Z. Kamalova studied racerunners and toad-head agamas living in the central region of the Fergana Valley [43,44,45].−T. Yadgarov studied snakes and the conservation of the endemic and unique ecosystems of the Fergana Valley [46,47]).−O.I. Tsaruk paid great attention to the problems and conservation of the unique ecosystems of the Fergana Valley, studying rare and endemic species of *Phrynocephalus* in the Fergana Valley [48].−Chikin (1998; 2001) assessed the status of populations of a number of species of rare and endemic reptiles in the sands of the Fergana Valley [49,50].−Bondarenko and Peregontsev (2017) described the spatial distribution of the Central Asian Tortoise, *Testudo horsfieldii* in Uzbekistan, including the Fergana Valley [51].−Currently, R.A. Nazarov and T.V. Abduraupov pay special attention to the study and conservation of the unique herpetofauna of the Fergana Valley, in particular, carrying out work on Papsky adyrs [4,52,53].

### 4.3. Conclusions

Lastly, the age of the Fergana Valley allows it to support subtle yet different habitats ranging from claystone outcroppings to massive, isolated sand massifs and xeric flats that each support their own suite of species that are locally adapted to these different micro-habitats. The discovery of possibly two new species (Figure 1 and Figure 3) that are micro-endemic habitat specialists isolated in various clay outcrops and canyons supports the hypothesis that the valley may have been colonized twice at different times. We hypothesize that this possible colonization pattern could indicate that other groups have experienced multiple colonization events and they too have levels of cryptic diversity among their populations. This study highlights the reptile diversity in the Fergana Valley and further supports the need for more surveys focusing on these habitats deemed “non-suitable”, because the data from this study indicate that they may harbor unrecognized biodiversity. If the biodiversity from these areas is not documented and protected, they could become victim to the continued textile and agricultural developments in the Fergana Valley.

## Figures and Tables

**Figure 1 animals-13-02516-f001:**
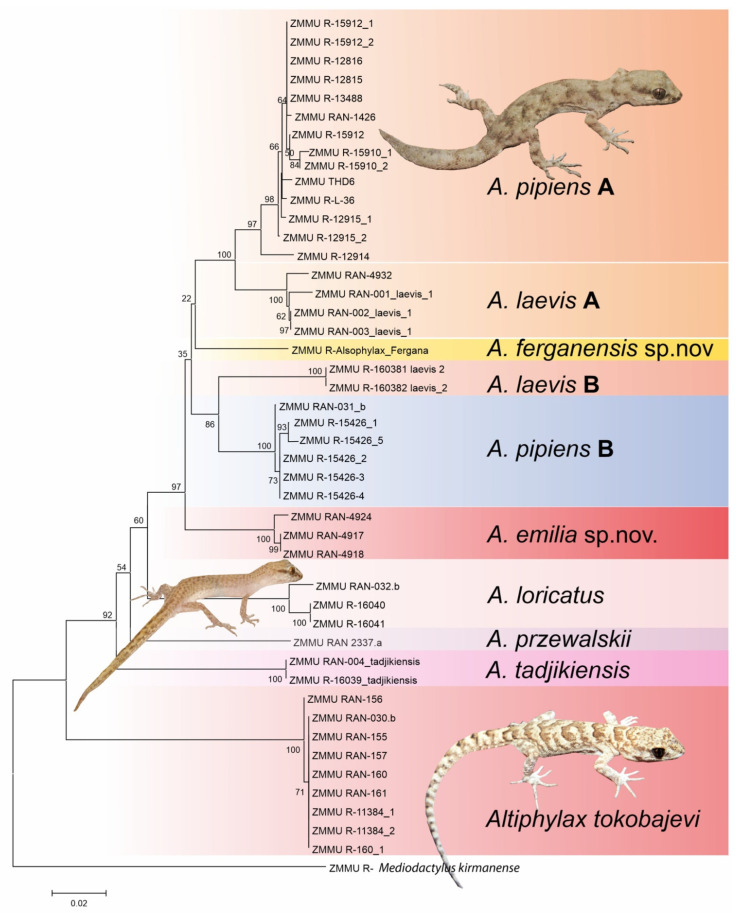
The gene tree from the maximum likelihood analysis for all described *Alsophylax* species using cytochrome oxidase I (COI).

**Figure 2 animals-13-02516-f002:**
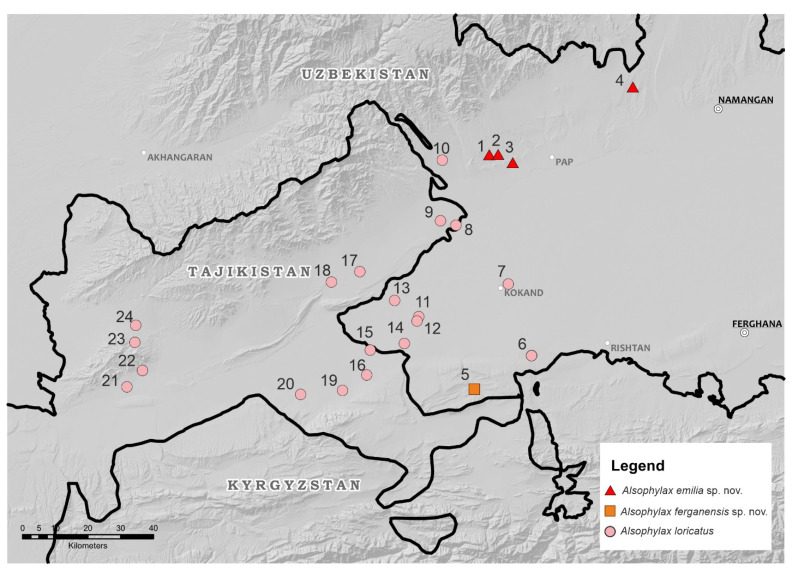
Distribution of three species of *Alsophylax* in the Fergana Valley.

**Figure 3 animals-13-02516-f003:**
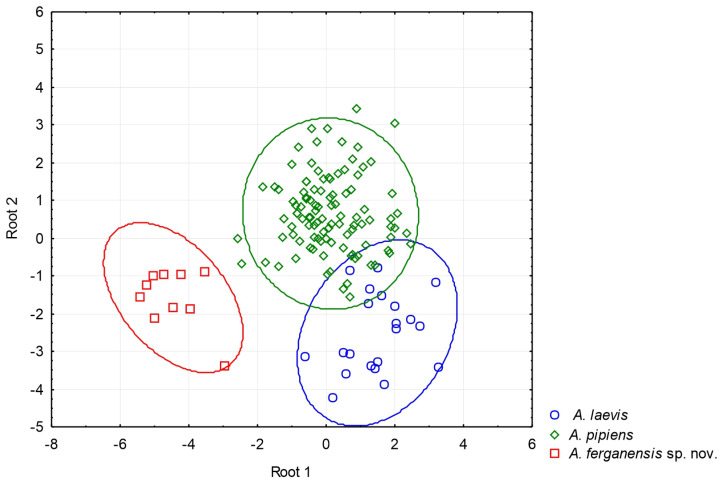
A plot from the discriminant function analysis for *Alsophylax laevis*, *A. pipiens*, and *A. ferganensis* sp. nov. The results of our discriminant function analysis demonstrate that *Alsophylax ferganensis* sp. nov. is morphometrically distinct from all clades of *A. laevis* and *A. pipiens*. Given the low sample size for *A. emlia* (n = 3), we were not able to make meaningful statistical comparisons with other species.

**Figure 4 animals-13-02516-f004:**
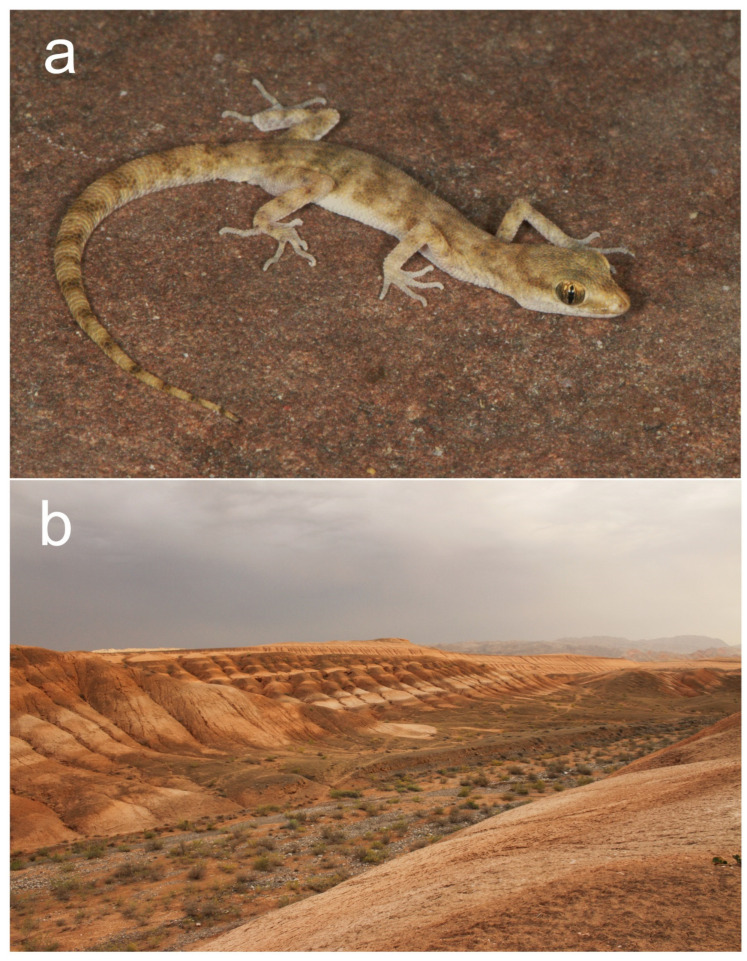
(**a**) *Alsophylax ferganensis* sp. nov. sp. nov. in situ and (**b**) its habitat.

**Figure 5 animals-13-02516-f005:**
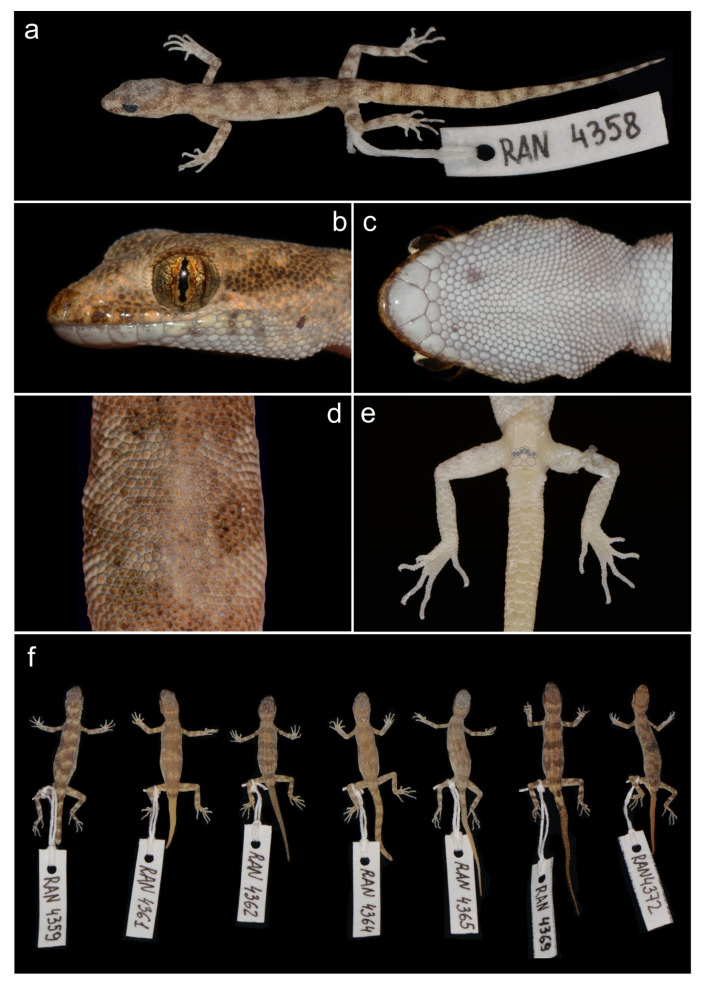
Holotype of *Alsophylax ferganensis* sp. nov. ZMMU Re-17532: (**a**) general view of preserved specimens; (**b**) lateral view of the head; (**c**) gular region showing mental, postmental, and chin scale arrangement; (**d**) middle of the back with homogeneous dorsal scales; (**e**) precloacal region showing pore and spur arrangements, and plantar view of feet showing subdigital lamellae; (**f**) general view of part of the paratype series of *A. ferganensis* sp. nov.

**Figure 6 animals-13-02516-f006:**
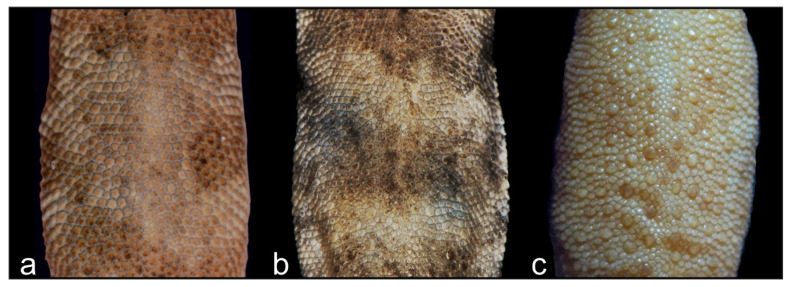
Three different types of dorsal scalation in *Alsophylax*: (**a**) *Alsophylax ferganensis* sp. nov. with homogeneous dorsal scales; (**b**) *Alsophylax emilia* sp. nov. with slightly heterogeneous dorsal scales and irregular enlarged tubercles; (**c**) *A. pipiens* with heterogeneous dorsal scales and regular rows of longitudinal enlarged tubercles.

**Figure 7 animals-13-02516-f007:**
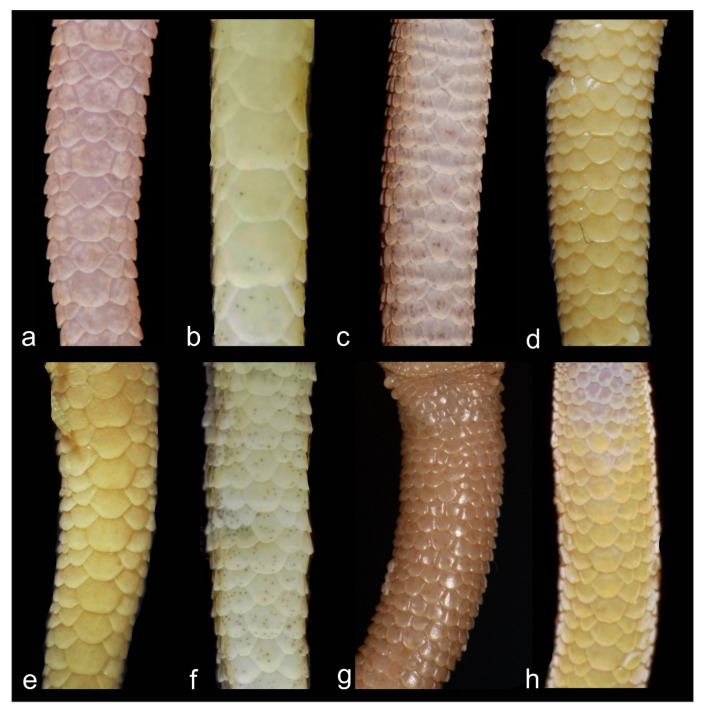
Different type of subcaudal scalation in *Alsophylax* species: (**a**) *A. ferganensis* sp. nov. with slightly enlarged medial scales; (**b**) *A. emilia* sp. nov. with medial row of transverse enlarged plates; (**c**) *A. pipiens* (clade B) from type locality (Mt. Bogdo) with approximately equal subcaudal scales; (**d**) *A. pipiens* (clade A) from Mongolia ZMMU R-5381, with slightly enlarged medial scales; (**e**) *A. pipiens* (clade B) form Ustyurt Plateau, western Uzbekistan ZMMU 15426, with noticeably enlarged medial scales; (**f**) *A. laevis* RAN 4957 from central Uzbekistan, Ayagitma depression; (**g**) *A. przewalskii* Paralectotypus ZISP 7016 with small homogeneous subcaudal scales; (**h**) *A. Tadjikiensis* ZMMU R-16039.

**Figure 8 animals-13-02516-f008:**
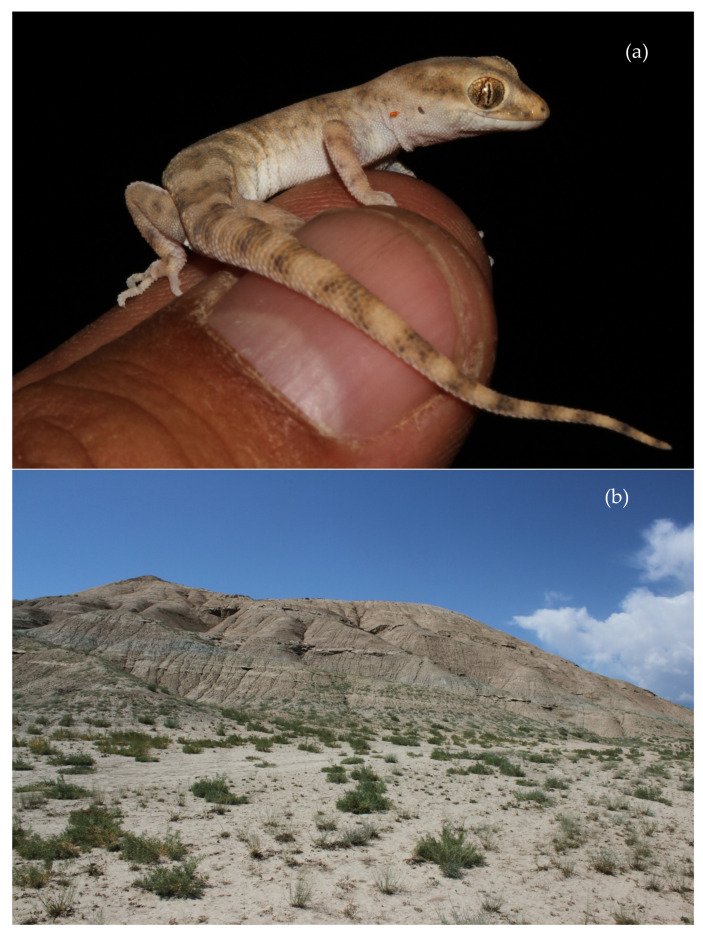
(**a**) *Alsophylax emilia* sp. nov. holotype and (**b**) its habitat.

**Figure 9 animals-13-02516-f009:**
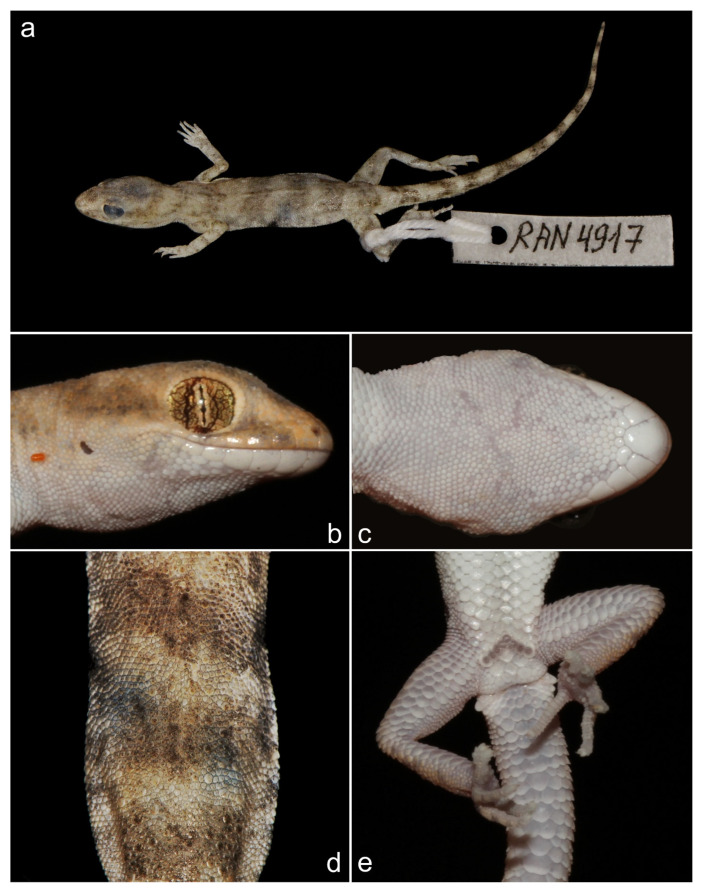
Holotype of *Alsophylax emilia* sp. nov. (ZMMU Re-17544): (**a**) general view of preserved specimen; (**b**) lateral view of the head; (**c**) gular region showing mental, postmental, and chin scale arrangement; (**d**) middle of the dorsum with heterogeneous dorsal scales; (**e**) precloacal region showing scalation pore and spur arrangements; (**e**) plantar view of feet showing subdigital lamellae.

**Figure 10 animals-13-02516-f010:**
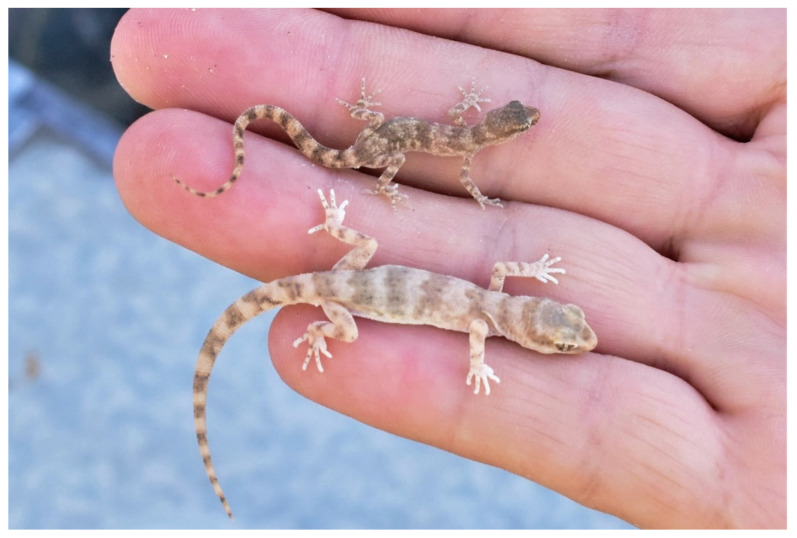
A comparison of adult male *Alsophylax ferganensis* sp. nov. (**top**) and *Alsophylax emilia* sp. nov. (**bottom**).

**Table 1 animals-13-02516-t001:** Density of rare and endemic retile species in key locations of the Fergana Valley.

Survey Point Number	*Alsophylax ferganensis* sp. nov.	*Alsophylax emilia* sp. nov.	*Alsophylax loricatus*	*Teratoscincus rustamowi* (Szczerbak, 1979)	*Phrynocephalus helioscopus* *saidalievi* (Sattorov, 1981)	*Phrynocephalus strauchi*	*Eremias* *scripta* *pherganensis*
1	5.25	-	-	-	0.1	-	-
2	-	-	11	-	-	-	-
3	-	-	-	54	-	6.8	7.7
4	-	-	-	25.3	-	3.2	6.8
5	-	-	-	58.5	-		9.6
6	-	-	-	36.3	-	2.4	2.7
7	-	-	-	-	2.3	-	-
8	-	3.4	-	-	3.4	-	-
9	-	4.5	-	-	4.2	-	-
10	-	-	-	-	3.7	-	-
11	-	-	-	-	4.1	1.6	-
12	-	-	-	14.7	-	3	-

1—Shorsu foothills (N40.24625; E70.81362); 2—fortress in Sary-Kurgan village (N40.334791; E71.021476); 3—Mingbulak Natural Monument (N40.77548; E71.35531); 4—Yazyavan Natural Monument (N40.716225; E71.487536); 5—sandy massif near Katta-Turk village (N40.63035; E70.81481); 6—sandy massif and clay uplands of Kairakum (N40.44350; E70.41715); 7—vicinity of the Altyaryk village (N40.355251; E71.591541); 8—pap foothills, 10 km northwest of Turakurgan village (N41.067757; E71.412176); 9—pap foothills, Jidalisai (N40.891349; E70.884367); 10—pap foothills, Uygursay (N40.95126; E71.01833); 11—pap foothills, 6 km south of Khanabad village (N40.831743; E70.736846); 12—vicinity of the Dangara village (N40.651748; E70.875879).

**Table 2 animals-13-02516-t002:** Measurements of the type series of *Alsophylax ferganensis* sp. nov.

	*Holotype*	*Paratypes*										
	ZMMU Re-17532	UZZI RE–19075	UZZI RE–19077	UZZI RE–19076	ZMMU Re-17533	UZZI RE–19078	ZMMU Re-17534	ZMMU Re-17535	ZMMU Re-17536	ZMMU Re-17537	ZMMU Re-17538	ZMMU Re-17539
*Field No.*	RAN 4358	RAN 4359	RAN 4360	RAN 4361	RAN 4362	RAN 4363	RAN 4364	RAN 4365	RAN 4369	RAN 4370	RAN 4371	RAN 4372
*SEX*	m	f	m	f	f	m	f	f	f	m	m	f
*SVL*	29.6	31.5	28.5	29.3	24.9	27.7	27.2	28.8	30.3	30	25.2	29
*TL*	36.9	-	29.6	-	-	-	-	28.7	32.9	37.7	32.1	23.7
*HL*	8.1	7.9	8	7.8	8	7.4	7.6	7.2	7.8	7.9	7	7.9
*HW*	4.6	4	4.2	3.9	3.5	4	4	3.8	4	3.9	3.5	3.7
*HH*	2.7	2.8	3	2.5	2.5	2.3	2.5	2.6	2.4	2.5	2	2
*SnEye*	3	2.7	2.7	2.6	2.3	2.7	2.5	2.4	2.5	2.5	2.4	2.4
*EyeEar*	2.7	2.2	2.4	2.6	2.1	2.1	2.4	2.2	2.4	2.5	2.1	2.2
*Orb D*	2	2	1.8	1.9	1.6	1.6	1.8	1.8	1.8	1.9	1.7	1.9
*LS*	4.7	4.3	4.6	4.5	4.4	4.4	4.6	4.6	4.7	4.3	4.2	4.5
*ForeaL*	4.3	4	4	3.8	3.5	3.8	3.9	4	4.1	3.8	3.5	3.9
*FemurL*	5.9	5.3	5.9	5.8	5.4	5.5	5.3	5.4	5.7	5.3	5.4	5.3
*CrusL*	5.9	5.6	5.6	5.5	4.9	5.1	5.2	5.1	6	5	5	5.6
*LD4A*	2.3	2.1	1.8	2.1	2.3	2.4	2.2	2.3	1.9	2.5	2.1	2.3
*LD4P*	3.1	3	2.7	3.1	3.2	2.9	3.1	3	2.8	3.3	2.9	3.3
*SL (r/l)*	7/7	7/7	7/7	7/7	7/7	7/7	7/7	7/8	7/8	7/7	8/6	7/7
*IL (r/l)*	5/5	5/6	5/5	5/5	5/5	5/5	5/5	6/6	5/6	5/5	6/5	5/5
*SEH*	18	18	22	21	20	19	18	20	19	19	20	19
*LF4*	14	14	14	15	15	13	13	15	13	13	12	14
*LT4*	18	15	16	15	14	15	15	17	16	14	15	18
*SLB*	86	87	86	83	89	88	89	97	85	84	94	94
*PP*	7	-	6	7	-	6	7	8	7	6	7	7
*Spurs (r/l)*	2/2	3/3	3/2	2/3	3/3	3/3	2/2	3/3	3/3	3/3	2/2	2/2
*Sq*	44	48	50	49	44	45	47	47	47	48	47	51

**Table 3 animals-13-02516-t003:** Morphological characteristics and data used in this study.

	*Alsophylax cf. laevis*	*Alsophylax laevis*	*Alsophylax loricatus*	*Alsophylax pipiens*	*Alsophylax cf. pipiens*	*Alsophylax przewalskii*	*Alsophylax szczerbaki*	*Alsophylax tadjikiensis*	*Alsophylax ferganensis*
*N*	21	60	25	46	107	19	17	3	12
SVL									
mean	31.97	32.55	30.74	33.14	32.00	30.35	28.26	27.10	28.50
±SD	1.65	2.34	3.02	3.06	2.47	1.58	1.27	2.07	1.50
range	27.2–35.2	26.6–38.7	24.6–40.9	27.4–41.5	26.2–39.8	26.59–33.5	26.3–32.4	24–29.1	24.9–31.5
TL									
mean	31.10	33.96	35.91	33.68	32.57	38.18	33.31	25.30	30.43
±SDSD	4.13	2.74	3.60	2.69	3.53	3.81	3.56	0.00	2.98
range	24–42.6	25–42	27.5–44.6	23.3–43.5	19–42.8	31.19–45.3	23.7–42.3	25.30	23.7–37.7
HL									
mean	8.86	8.87	8.32	9.07	8.89	8.37	7.98	7.53	7.72
±SDSD	0.44	0.55	0.91	0.79	0.63	0.32	0.30	0.49	0.28
range	7.9–9.7	7.4–10.3	7–10.7	7.5–10.9	7.2–11	7.02–8.93	7.3–8.8	6.8–8	7–8.1
HW									
mean	5.02	4.94	4.44	5.12	5.14	4.82	4.24	3.90	3.93
±SD	0.32	0.36	0.48	0.52	0.45	0.38	0.20	0.13	0.21
range	4.4–5.7	3.9–5.9	3.6–5.7	3.6–6.7	4–6.5	4.2–5.8	3.7–4.7	3.7–3.9	3.5–4.6
HH									
mean	3.18	3.06	2.70	3.31	3.32	3.36	2.69	2.50	2.48
±SD	0.32	0.28	0.32	0.35	0.36	0.41	0.16	0.13	0.21
range	2.3–4.2	2.4–4.2	2–3.51	2.2–4.1	2.2–4.5	2.5–4.33	2.2–3	2.7–2.4	2–3
SnEye									
mean	2.92	2.86	2.80	2.98	2.91	2.74	2.52	2.43	2.56
±SD	0.22	0.18	0.30	0.30	0.22	0.15	0.09	0.09	0.15
range	2.5–3.4	2.3–3.4	2.3–3.6	2.4–3.8	2.4–3.6	2.19–3	2.3–2.7	2.3–2.5	2.3–3
Orb D									
mean	2.18	2.05	1.82	2.15	2.15	1.72	1.74	1.80	1.82
±SD	0.15	0.11	0.29	0.18	0.17	0.23	0.08	0.20	0.10
range	1.9–2.6	1.6–2.5	1.26–2.7	1.69–2.8	1.6–2.7	1.18–2	1.5–1.9	1.5–2.1	1.6–2
LS									
mean	5.11	4.70	4.40	4.93	4.72	4.39	3.83	3.93	4.48
±SD	0.40	0.31	0.70	0.53	0.40	0.33	0.13	0.16	0.14
range	4.2–6.7	3.6–5.3	3.5–6	3.8–6.2	3.6–6.4	3.52–5	3.5–4.2	3.7–4	4.2–4.7
ForeaL									
mean	4.40	3.85	3.49	4.15	4.00	3.51	3.15	3.37	3.88
±SD	0.29	0.26	0.59	0.45	0.31	0.27	0.14	0.18	0.17
range	3.8–5.6	3–4.7	2.7–5	3–5.5	2.9–5	3.1–4.11	3–3.4	3.1–3.4	3.5–4.3
FemurL									
mean	5.89	5.29	5.19	5.91	5.74	5.17	4.45	4.53	5.52
±SD	0.35	0.32	0.74	0.65	0.44	0.29	0.22	0.11	0.21
range	5.2–6.6	4.2–6.1	4–7.2	4.4–7.9	4.5–7.3	4.5–5.71	4.1–5.2	4.7–4.4	5.3–5.9
CrusL									
mean	5.70	5.22	4.73	5.60	5.27	4.84	4.31	4.50	5.38
±SD	0.34	0.36	0.80	0.57	0.48	0.34	0.18	0.07	0.33
range	5–6.5	4–6	3.6–6.8	4.3–7.1	3.9–6.7	4.1–5.49	3.9–4.7	4.4–4.5	4.9–6
LD4A									
mean	2.20	2.21	2.52	2.76	2.81	2.50	2.26	1.90	2.19
±SD	0.17	0.17	0.40	0.31	0.26	0.18	0.21	0.07	0.16
range	1.9–2.6	1.8–2.7	1.8–3.6	2.1–3.6	2–3.5	2.1–2.9	1.5–2.8	1.9–1.8	1.8–2.5
LD4P									
mean	2.97	2.92	3.45	3.67	3.58	3.65	3.19	2.33	3.03
±SD	0.24	0.19	0.39	0.38	0.33	0.25	0.16	0.18	0.15
range	2.2–3.7	2.3–3.4	2.1–4.4	2.7–4.41	2.7–4.4	2.75–4.17	2.9–3.6	2.2–2.2	2.7–3.3
EyeEar									
mean	2.61	2.49	2.46	2.70	2.65	2.48	2.55	2.47	2.33
±SD	0.23	0.21	0.23	0.28	0.25	0.17	0.14	0.11	0.18
range	2–3	2–3	2–3	2.1–3.4	1.9–3.6	2.2–2.89	2.3–2.8	2.3–2.5	2.1–2.7

**Table 4 animals-13-02516-t004:** Measurements of the type series of *Alsophylax emilia* sp. nov.

	*Holotype*	*Paratypes*	
	ZMMU Re-17544	UZZI RE–19079	ZMMU Re-17545
*Field No.*	RAN 4917	RAN 4918	RAN 4924
*SEX*	m	m	m
*SVL*	35	32.6	31.5
*TL*	40.2	15.3 regenerated	33.1
*HL*	9.7	9.2	9
*HW*	5.8	5.2	5.1
*HH*	2.7	2.8	2.6
*SnEye*	3	2.8	3
*EyeEar*	2.9	2.8	2.6
*Orb D*	2.1	2.2	2.2
*LS*	5.8	5.4	5.2
*ForeaL*	5.1	4.7	4.6
*FemurL*	6.5	6.4	5.3
*CrusL*	6.5	6.0	5.7
*LD4A*	2.5	2.2	2.2
*LD4P*	3	2.6	2.6
*SL (r/l)*	8/8	7/7	8/9
*IL (r/l)*	6/7	6/6	7/6
*SEH*	22	22	22
*LF4*	11	12	11
*LT4*	13	14	14
*SLB*	97	102	104
*PP*	8	9	7
*Spurs (r/l)*	2/2	2/3	2/2
*Sq*	65	64	59

## Data Availability

All data were deposited online.

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
