# Peer review of "The Fergana Valley Is an Isolate of Biodiversity: A Discussion of the Endemic Herpetofauna and Description of Two New Species of Alsophylax (Sauria: Gekkonidae) from Eastern Uzbekistan"

_animals, 2023, doi:10.3390/ani13152516_

Round 1
Reviewer 1 Report
In this study, the authors obtained up-to-date data on the distribution and abundance of five endemic reptile species in the remaining isolated and undeveloped habitats across the Fergana Valley in Uzbekistan and discovered two unique and new micro-endemic species of gecko. I think this work is really meaningful. As we all know, animal protection work in developing countries always faces some difficulties, such as limited funds and difficulties in investigation. The authors have engaged in an effective international collaboration and have achieved an appreciable result. However, the paper contains many errors in some details, and my comments are as follows.
1. There are 10 Figures in the article, but not all of them are mentioned in the text, e.g. Figure 8, please check the manuscript carefully.
2. Similarly, there are 4 tables in the manuscript, but only Table 1 is mentioned in the article.
3. The format of this article does not match the journal's author guidelines, such as figure line 128, (SFig.1), line 558 and line 309, and citation of references. Please read the journal's author guidelines carefully.
4. Author Contributions missing.
5. In Figure 1, there were two ZMMU R-15426_1 and two ZMMU R-15426-5. This is quite confusing. Does this mean different transcripts of the same gene? Even if it is different transcripts of the same gene, the differences are too great, please check it.
6. The legend of Figure 1 is too simple and lacks a lot of information and explanation.
7. Reasons for selecting the COI gene should be given in the article.
8. The Latin name in Table 1 is the full name, and the Latin name in Table 2 is abbreviated and not italicized.
9. Table 3 and 4 is difficult to understand due to a lack of explanation, and some abbreviations in the Table are not given full names in the article.
10. In the summary section, the authors mentioned the urgent need to create state protected areas of habitat with IUCN I and II protection status for the remaining areas of suitable habitat. However, this was not mentioned in the discussion, which I think is important.
11. The authors' results are well presented, but the discussion is a little superficial.
Author Response
Reviewer 1
In this study, the authors obtained up-to-date data on the distribution and abundance of five endemic reptile species in the remaining isolated and undeveloped habitats across the Fergana Valley in Uzbekistan and discovered two unique and new micro-endemic species of gecko. I think this work is really meaningful. As we all know, animal protection work in developing countries always faces some difficulties, such as limited funds and difficulties in investigation. The authors have engaged in an effective international collaboration and have achieved an appreciable result. However, the paper contains many errors in some details, and my comments are as follows.
- There are 10 Figures in the article, but not all of them are mentioned in the text, e.g. Figure 8, please check the manuscript carefully. We fixed this is the MS
- Similarly, there are 4 tables in the manuscript, but only Table 1 is mentioned in the article. We fixed this is the MS
- The format of this article does not match the journal's author guidelines, such as figure line 128, (SFig.1), line 558 and line 309, and citation of references. Please read the journal's author guidelines carefully. We fixed this is figure legend.
- Author Contributions missing. We fixed this is the MS
- In Figure 1, there were two ZMMU R-15426_1 and two ZMMU R-15426-5. This is quite confusing. Does this mean different transcripts of the same gene? Even if it is different transcripts of the same gene, the differences are too great, please check it. We fixed this is the figure.
- The legend of Figure 1 is too simple and lacks a lot of information and explanation. We feel the figure legend is sufficient in explanation.
- Reasons for selecting the COI gene should be given in the article. We did.
- The Latin name in Table 1 is the full name, and the Latin name in Table 2 is abbreviated and not italicized. We fixed this is the MS.
- Table 3 and 4 is difficult to understand due to a lack of explanation, and some abbreviations in the Table are not given full names in the article. We fixed this is the MS
- In the summary section, the authors mentioned the urgent need to create state protected areas of habitat with IUCN I and II protection status for the remaining areas of suitable habitat. However, this was not mentioned in the discussion, which I think is important. We did mention this.
- The authors' results are well presented, but the discussion is a little superficial. This an opinion and not an edit.
Reviewer 2 Report
This manuscript identifies and describes two new species of the poorly studied miniaturized gekkonid genus Alsophylax from the Fergana Valley in Uzbekistan. The systematic part of the work is placed into the context of reptile endemism in the Fergana Valley.
The descriptions and diagnoses are well-executed and the data provided for morphology and genetics support the recognition of the new species. Overall the paper accomplishes its goals, but there are several issues that require attention.
The sections of the paper discussing reptile endemism are more superficial and should be expanded in a dedicated paper. In Table 1 the units of density are not given.
Although I am convinced of the distinctiveness of the new taxa, and CO1 is perhaps sufficient to demonstrate that, it is notoriously poor, on its own, as the basis for phylogenetic relationships. The combination of the very weak support for many nodes, including certain key nodes, like those relating to the placement of A. ferganensis, makes any phylogenetic inference suspect. For example, the lack of support suggests that A. ferganensis could be sister to A. pipiens + A. laevis and that A. laevis might be monophyletic, in contrast to the assertion in the text. Further the reference at several points in the manuscript to two groups with differing dorsal scale heterogeneity in the genus needs to be qualified, as readers might think these are monophyletic groups, but based on this tree they are not. This is especially a problem in the discussion as the unsupported claim is used to argue for multiple colonizations of the Fergana Valley. It is also unclear why the authors provide methods for parsimony and maximum likelihood, when only a Bayesian tree is presented and it is also unclear why the figure legend for the tree indicates that the numbers represent posterior probabilities/bootstrap values, when only one value (presumably bootstrap) is depicted on the tree.
With respect to the morphological methods and descriptions it is unclear why the upper arm is called the "shoulder." This should be "upper arm", "humeral length" or "propodial length". Throughout the manuscript scales are refered to as "tiled", presumably this means "juxtaposed", which is the appropriate term for scales like this. On line 301 Mensural would be a better term than morphometric. I also noted one obvious inconsistency: the maximum length of A. ferganensis is variously given as 31.7 (line 347), 31.5 (line 339) and 30.3 (line 544).
In the material examined, under A. pipiens, the locality "Mongolia, neighborhood 5 plot profile" is meaningless to the reader. Unless some locatable place name is used this should simply say "Mongolia".
It is stated that all data are deposited online, but no link or indication of repository is provided. As this is an electronic journal, the new names will require ZooBank registration. Presumably this will be done at a later point in processing, but I flag it here as a precaution.
The manuscript is fully comprehensible but has an inordinate amount of grammatical errors, chiefly involving mismatch in number (e.g. a plural noun and singular verb), or the choice of inappropriate words. Given that there are native English-speaking authors this is really unacceptable.
line 6 comma after Lapin
33 comma after development
69 comma before Fergana
70 "and" befor Strauch's
71 massif not massifs
72 replace "are" with "comprise"
73 have hot has
78 greatest not strongest
79 farming not farms; vegetable not vegetables
83 led not lead; add "Being" after habitats
85 disrupted not broken
93 massif not massifs
108 has not have
116 gekkonid
121 distribution oif the genus
124 new not news
125 comprises not "consists of"
145 along not at
156 shift equations to right
159 antecedent of "its" unclear
168 change the to their
196 satisfactory not satisfying
238 include not includes
246 axilla not auxillary
259 base not basis
289 delete "of"
293 add "to" after "sister"
310 among should be with
317 size should be length
319 are not is
321 italics needed
335 unclear what is meant by "prizes"
342 are not is
352 in not on
367-68 move "ventral" to immediately before the word "scale"
383 sole not foot
402 unclear what 50 km2 is a mesasurement of
409 ephemera is incorrect word
424 italicize name (and presumably leave a gap to separate it from the previous species)
440 larger not large
444 semicolon before 8/8
450 change bigger to larger
462 "prizes" see comment above
464 postnasal not postnasals
470-471 awkward, reword
472 forming not formed; larger not large
489 this is not a sentence
493 transverse not transversal
494 somewhat narrower
518 are not is
521 across not among; organized not organize
522 in not on mentioned not mention
546 see line 493
552 delete "Two"
567 work not works
574 Lichtenstein's first name was Martin, but he went by Hinrich [presumably the "g" here is because the same letter in Russian can be "G" or "H".
576 Lichtenstein misspelled\582 delete period, lower case"e" on expanded
586 "which are described"
597 gave not was
616 add "the" before Fergana
622 should read "support subtle"
624 suite not sweet
625 isolated not isolate
627 colonized not colonize
667 Helminthofauna [again a transliteration issue from Russian]
670 Miocene (capital)
725 Lichtenstein, M.H.C. (only one author)
736 capitals needed for the higher order scientific group names
745 Russo-Asiatica
767 Italics on Phrynocephalus
Author Response
Reviewer 2
This manuscript identifies and describes two new species of the poorly studied miniaturized gekkonid genus Alsophylax from the Fergana Valley in Uzbekistan. The systematic part of the work is placed into the context of reptile endemism in the Fergana Valley.
The descriptions and diagnoses are well-executed and the data provided for morphology and genetics support the recognition of the new species. Overall the paper accomplishes its goals, but there are several issues that require attention.
The sections of the paper discussing reptile endemism are more superficial and should be expanded in a dedicated paper. In Table 1 the units of density are not given. Agreeded.
Although I am convinced of the distinctiveness of the new taxa, and CO1 is perhaps sufficient to demonstrate that, it is notoriously poor, on its own, as the basis for phylogenetic relationships. The combination of the very weak support for many nodes, including certain key nodes, like those relating to the placement of A. ferganensis, makes any phylogenetic inference suspect. For example, the lack of support suggests that A. ferganensis could be sister to A. pipiens + A. laevis and that A. laevis might be monophyletic, in contrast to the assertion in the text. Further the reference at several points in the manuscript to two groups with differing dorsal scale heterogeneity in the genus needs to be qualified, as readers might think these are monophyletic groups, but based on this tree they are not. This is especially a problem in the discussion as the unsupported claim is used to argue for multiple colonizations of the Fergana Valley. It is also unclear why the authors provide methods for parsimony and maximum likelihood, when only a Bayesian tree is presented and it is also unclear why the figure legend for the tree indicates that the numbers represent posterior probabilities/bootstrap values, when only one value (presumably bootstrap) is depicted on the tree. We fixed this is the Methods section.
With respect to the morphological methods and descriptions it is unclear why the upper arm is called the "shoulder." This should be "upper arm", "humeral length" or "propodial length". Throughout the manuscript scales are refered to as "tiled", presumably this means "juxtaposed", which is the appropriate term for scales like this. On line 301 Mensural would be a better term than morphometric. I also noted one obvious inconsistency: the maximum length of A. ferganensis is variously given as 31.7 (line 347), 31.5 (line 339) and 30.3 (line 544). We fixed this is the MS.
In the material examined, under A. pipiens, the locality "Mongolia, neighborhood 5 plot profile" is meaningless to the reader. Unless some locatable place name is used this should simply say "Mongolia". We fixed this is the MS.
It is stated that all data are deposited online, but no link or indication of repository is provided. As this is an electronic journal, the new names will require ZooBank registration. Presumably this will be done at a later point in processing, but I flag it here as a precaution. We say in the MS and we fixed this is the MS.
Comments on the Quality of English Language
The manuscript is fully comprehensible but has an inordinate amount of grammatical errors, chiefly involving mismatch in number (e.g. a plural noun and singular verb), or the choice of inappropriate words. Given that there are native English-speaking authors this is really unacceptable. Agreed.
line 6 comma after Lapin. We made all the following edits.
33 comma after development
69 comma before Fergana
70 "and" befor Strauch's
71 massif not massifs
72 replace "are" with "comprise"
73 have hot has
78 greatest not strongest
79 farming not farms; vegetable not vegetables
83 led not lead; add "Being" after habitats
85 disrupted not broken
93 massif not massifs
108 has not have
116 gekkonid
121 distribution oif the genus
124 new not news
125 comprises not "consists of"
145 along not at
156 shift equations to right
159 antecedent of "its" unclear
168 change the to their
196 satisfactory not satisfying
238 include not includes
246 axilla not auxillary
259 base not basis
289 delete "of"
293 add "to" after "sister"
310 among should be with
317 size should be length
319 are not is
321 italics needed
335 unclear what is meant by "prizes"
342 are not is
352 in not on
367-68 move "ventral" to immediately before the word "scale"
383 sole not foot
402 unclear what 50 km2 is a mesasurement of
409 ephemera is incorrect word
424 italicize name (and presumably leave a gap to separate it from the previous species)
440 larger not large
444 semicolon before 8/8
450 change bigger to larger
462 "prizes" see comment above
464 postnasal not postnasals
470-471 awkward, reword
472 forming not formed; larger not large
489 this is not a sentence
493 transverse not transversal
494 somewhat narrower
518 are not is
521 across not among; organized not organize
522 in not on mentioned not mention
546 see line 493
552 delete "Two"
567 work not works
574 Lichtenstein's first name was Martin, but he went by Hinrich [presumably the "g" here is because the same letter in Russian can be "G" or "H".
576 Lichtenstein misspelled\582 delete period, lower case"e" on expanded
586 "which are described"
597 gave not was
616 add "the" before Fergana
622 should read "support subtle"
624 suite not sweet
625 isolated not isolate
627 colonized not colonize
667 Helminthofauna [again a transliteration issue from Russian]
670 Miocene (capital)
725 Lichtenstein, M.H.C. (only one author)
736 capitals needed for the higher order scientific group names
745 Russo-Asiatica
767 Italics on Phrynocephalus
Reviewer 3 Report
The ms is of great interest in globale sense of study biodiversity. Taxonomic part if also very high quality.
The ms should be published after very few corrections.
Table 2. Morphometric characters and data used in this study - why A.tadjikiensis is marked by RED. Where the authors positioned A.emilia? As A.cf.pipiens? These ponts required explanations.
Table 1. retile - misprint
Author Response
Reviewer 3
The ms is of great interest in globale sense of study biodiversity. Taxonomic part if also very high quality. Thank you.
The ms should be published after very few corrections.
Table 2. Morphometric characters and data used in this study - why A.tadjikiensis is marked by RED. Where the authors positioned A.emilia? As A.cf.pipiens? These ponts required explanations. We fixed this is the MS.
Table 1. retile – misprint. We fixed this is the MS.